# Lie-Equivariant Quantum Graph Neural Networks

**Jogi Suda Neto**
QuaTI,
13560-161 São Carlos, SP, Brazil
`jogi.suda@unesp.br`

**Roy T. Forestano**
Physics Department,
University of Florida
Gainesville, FL 32611
`roy.forestano@ufl.edu`

**Sergei Gleyzer**
Dep. Physics & Astronomy
University of Alabama
Tuscaloosa, AL 35487
`sgleyzer@ua.edu`

**Kyoungchul Kong**
Dep. Physics & Astronomy
University of Kansas
Lawrence, KS 66045
`kckong@ku.edu`

**Konstantin T. Matchev**
Physics Department,
University of Florida
Gainesville, FL 32611
`matchev@ufl.edu`

**Katia Matcheva**
Physics Department,
University of Florida
Gainesville, FL 32611
`matcheva@ufl.edu`

## Abstract

Discovering new phenomena at the Large Hadron Collider (LHC) involves the identification of rare signals over conventional backgrounds. Thus binary classification tasks are ubiquitous in analyses of the vast amounts of LHC data. We develop a Lie-Equivariant Quantum Graph Neural Network (Lie-EQGNN), a quantum model that is not only data efficient, but also has symmetry-preserving properties. Since Lorentz group equivariance has been shown to be beneficial for jet tagging, we build a Lorentz-equivariant quantum GNN for quark-gluon jet discrimination and show that its performance is on par with its classical state-of-the-art counterpart LorentzNet, making it a viable alternative to the conventional computing paradigm.

## 1 Introduction

The onset of the high luminosity stage of the Large Hadron Collider (LHC) [1] by the end of this decade presents a formidable computational challenge to effectively manage and analyze the resulting datasets [2]. A promising new computing paradigm for tackling this problem is the application of quantum machine learning (QML), which offers the possibility of reducing the time complexity of classical algorithms by running on quantum computers, which can have access to the exponentially large Hilbert space [3–9].

In particle physics, symmetries underpin the fundamental laws governing the interactions and behaviors of subatomic particles: the Standard Model (SM), for example, is built upon symmetry principles such as gauge and Lorentz invariance, which dictate the interactions between particles and fields [10]. At the same time, symmetries play a crucial role in machine learning as well — if a neural network is invariant under some group, then its output could be expressed as functions of the orbits of the group [11]. In recent years, the field of *Geometric Deep Learning* [12] was born, and it posits that despite the curse of dimensionality, the success of deep learning can be explained by two main inductive biases: symmetry and scale separation. By leveraging inherent symmetries in the data, the hypothesis space is effectively reduced, models can become more compact, and require less data for training - attributes that are particularly critical for quantum computing, given the current hardware limitations.

Recognizing the intertwined nature of symmetries and deep learning, a plethora of models have been developed that are invariant to different groups. Convolutional neural networks (CNNs), for example, which have revolutionized computer vision, are naturally invariant to image translations. Transformer-

38th Second Workshop on Machine Learning with New Compute Paradigms at NeurIPS 2024(MLNCP 2024).

based language models, on the other hand, exhibit permutation invariance. These architectures, when provided with appropriate data, are capable of learning stable latent representations under the action of their respective groups. An interesting question to be explored in quantum machine learning is the use of symmetries [13–17], especially in the context of particle physics [18–22].

Since the data generated by collision events is usually represented in a graph format, graph neural networks (GNNs) [23] are a natural choice in particle physics [24]. Furthermore, Lorentz symmetry is a fundamental spacetime symmetry for any relativistic model of elementary particles. The state of the art implementation of Lorentz symmetry in a GNN is LorentzNet [25], which has been used for jet tagging, i.e., identifying the progenitor elementary particle which initiates the jet. In this paper we focus on a specific version of the jet tagging problem, namely, discriminating between light quark and gluon jets [26–28]. We build a Lorentz-equivariant quantum GNN for quark-gluon jet discrimination and show that its performance is on par with its classical state-of-the-art counterpart, LorentzNet. This is an encouraging result, given that quantum computation is still in its infancy stage. In the NISQ era, our focus is on quantum utility, i.e., effectiveness and practicality of quantum computers in terms of speed, accuracy or energy consumption, over a classical machine of similar size, weight, and cost [29].

## 2 Lie-Equivariance

Given the abstract definition of Lie groups as used in particle physics [30, 31], our primary goal is to understand how these symmetries can be embedded in hybrid quantum-classical ML architectures. Mathematically, we consider our data as a set of vectors $x$ sampled from an input space $\mathcal{X} = \mathbb{R}^n$. We then describe how a group element $g \in G$ acts linearly on $x$ through $T_{\mathcal{X}}(g)$, where $T_{\mathcal{X}} : G \to \mathbb{GL}(n)$ is known as the group representation. Essentially, $T_{\mathcal{X}}$ maps each element of the group $G$ to a nonsingular matrix $T_{\mathcal{X}}(g) \in \mathbb{R}^{n \times n}$. It is important to note that each representation of a Lie group induces a corresponding representation in its Lie algebra.

The idea of symmetry-preserving behavior in machine learning is essential. By reducing the hypothesis space, the learning process becomes more efficient. Additionally, models that integrate geometric information as an inductive bias tend to be naturally data-efficient, eliminating the need for data augmentation during training [13–17, 25, 32–35]. This is particularly important in Quantum Machine Learning (QML), where the limited number of qubits makes data efficiency crucial [18, 19].

For a given function $f : x \in \mathcal{X} \to y \in \mathcal{Y}$ (for example, $f$ could be a neural network, $\mathcal{X}$ the input space, and $\mathcal{Y}$ a latent space) and some group $G$ that acts on both $\mathcal{X}$ and $\mathcal{Y}$, the $f$ holds the desirable property of equivariance if $\forall g \in G, (x, y) \in \mathcal{D}, T_y(g)y = f(T_x(g)x)$. For brevity of notation, we write it as: $g \cdot y = f(g \cdot x)$.

It becomes clear that the invariance is a special case of equivariance where $f(g \cdot x) = f(x)$, that is, $T_y(g) = \mathbb{I}$, so every group element acting in the space of $\mathcal{Y}$ gets mapped to the identity. To incorporate invariance into the architectures, it is essential to first identify the group $G$ that accounts for the underlying transformations of the data. In our context, these groups are Lie groups. In particular we will consider the Lorentz group, which is a fundamental symmetry of the standard model of particle physics. Our objective is to build a Quantum Algorithm for Lorentz-Equivariant Graph Neural Network for gluon/quark discrimination.

## 3 Dataset

In this work, we consider the task of determining whether a given jet originated from a quark or a gluon (this is known as jet-tagging). For illustration we use the high energy physics dataset *Pythia8 Quark and Gluon Jets for Energy Flow* [36], which contains two million jets split equally into one million quark jets and one million gluon jets. These jets resulted from simulated LHC collisions with total center of mass energy $\sqrt{s} = 14$ TeV and were selected to have transverse momenta $p_T^{jet}$ between 500 to 550 GeV and rapidities $|y^{jet}| < 1.7$. For our analysis, we randomly picked $N = 12500$ jets and used the first 10000 for training, the next 1250 for validation, and the last 1250 for testing. These sets happened to contain 4982, 658, and 583 quark jets, respectively.

For our purposes, we consider the jet dataset to be constituted of point-clouds, where each jet is represented as a graph $\mathcal{G} = \{\mathcal{V}, \mathcal{E}\}$, i.e., a set of nodes and edges, respectively. (This is the natural

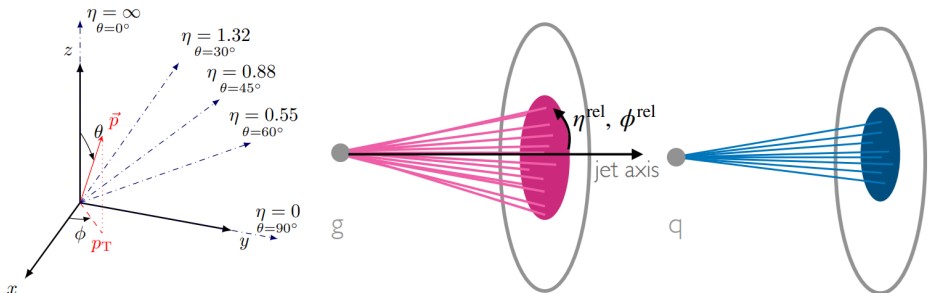

Figure 1: The coordinate system (left) used to represent components of the particle momentum $\vec{p}$ (see also Fig. 1 in [21]). Schematic representation of a gluon jet (middle) and a quark jet (right). The jet constituents (solid lines) are collimated around the jet axis. (Figure adapted from Fig. 1 in [37].)

data structure used by Graph Neural Networks.) In our case, each node has the transverse momentum $p_T$, pseudorapidity $\eta$, azimuthal angle $\phi$ (and other scalar-like quantities like particle ID, particle mass, etc.) of the respective constituent particle in the jet (see Figure 1). Generally, the number of features is always constant, but the number of nodes in each jet may vary. For our analysis we used jets with at least 10 particles.

## 4 Models

### 4.1 Graph Neural Networks

Graph Neural Networks are a class of neural networks designed to operate on graph-structured data. Unlike traditional neural networks that work on Euclidean data such as images or sequences, GNNs are capable of capturing the dependencies and relationships between nodes in a graph. The fundamental idea behind GNNs is to iteratively update the representation of each node by aggregating information from its neighbors, thereby enabling the network to learn both local and global graph structures.

Mathematically, a GNN can be described as follows: Let $G = (V, E)$ be a graph where $V$ is the set of nodes and $E$ is the set of edges. Each node $v \in V$ has an initial feature vector $\mathbf{h}_v^{(0)}$. The node features are updated through multiple layers of the GNN. At each layer $l$, the feature vector of node $v$ is updated by aggregating the feature vectors of its neighbors $\mathcal{N}(v)$ and combining them with its own feature vector. This can be expressed as:

$$\mathbf{h}_v^{(l+1)} = \sigma\Big(\mathbf{W}^{(l)}\mathbf{h}_v^{(l)} + \sum_{u \in \mathcal{N}(v)} \mathbf{W}_e^{(l)}\mathbf{h}_u^{(l)}\Big)$$

where $\mathbf{W}^{(l)}$ and $\mathbf{W}_e^{(l)}$ are learnable weight matrices, $\sigma$ is a non-linear activation function, and $\mathcal{N}(v)$ denotes the set of neighbors of node $v$. This process is repeated for a fixed number of layers, allowing the network to capture higher-order neighborhood information. The final node representations can be used for various downstream tasks such as node classification, link prediction, or graph classification.

### 4.2 Equivariant Graph Neural Networks

As a simple example of equivariance in the context of classical GNNs, consider a dataset of graphs in which the nodes have cartesian coordinates as input features, and let $SO(2)$ be their underlying group. Since the Euclidean norm - induced by the euclidean metric - is invariant to rotations, any node-updating function of the form:

$$x_i^{l+1} = x_i^l + C \sum_{j \in \mathcal{N}(i)} (x_i^l - x_j^l)\, \phi_x(m_{ij}^l) \tag{1}$$

is naturally equivariant (for the proof, see Appendix A in [19]). Here $\phi_x$ is some classical neural network and $m_{ij}^l$ is the *edge message* between nodes $i$ and $j$ in layer $l$. See Ref. [25] for different classical neural networks, $\phi_e$, $\phi_m$ and $\phi_h$.

Equivariant GNNs (EGNNs) can be applied to other groups of interests, like the Lorentz group, a central piece in high energy physics [25].

Particles observed at the LHC are moving at velocities comparable to the speed of light, and are therefore described by special relativity. Mathematically, the transformations between two inertial frames of reference are described by the Lorentz group $O(1,3)$.

## 4.3 Quantum Graph Neural Networks

Having established that classical GNNs aim to learn a lower-dimensional, topologically preserving representation of graphs, we now explore quantum graph neural networks (QGNNs). One approach for QGNNs involves encoding the graph's nodes and edges into any Hamiltonian whose topology of interactions is that of the problem graph [38]. Previous works [19, 38] have utilized the transverse-field Ising model, initially introduced to study phase transitions in magnetic systems. Once the graph is encoded in the Hamiltonian, it is transformed into a quantum circuit using a Trotter-Suzuki approximation [39].

This method naturally yields a permutation-equivariant QGNN via the Ising Hamiltonian. As discussed, the symmetries from quantum fields, describing the invariance of physical laws under different inertial frames, are represented by the Lorentz group. Prior research outlines efficient strategies for achieving equivariance. For instance, [40] demonstrates that an invariant model can be constructed using an equivariant set of quantum gates. However, this framework fundamentally assumes the presence of either discrete or compact Lie groups.

Given that the Lorentz group is noncompact, here we adopt an alternative method based on Invariant Theory [41, 42], which is different from the strategy to achieve equivariance explained before [40]. In addition, our approach for QGNNs is different from the one used in [19, 38]. It has previously been applied in the classical setting for High-Energy Physics [25], and is the main motivation for our work.

## 4.4 Lie-Equivariant Quantum Graph Neural Network

Similarly to LorentzNet, for standard jet tagging approach, our input is made of 4-momentum vectors and any associated particle scalar one may wish to include, like color and charge. We start with the traditional LorentzNet architecture (Fig. 1 in Ref. [25]), with the following modifications: $\phi_e$, $\phi_x$, $\phi_h$, and $\phi_m$, i.e., the classical multilayer perceptrons in LorentzNet, are substituted by quantum counterparts, that is, variational circuits, whose architecture is illustrated below in Fig. 2:

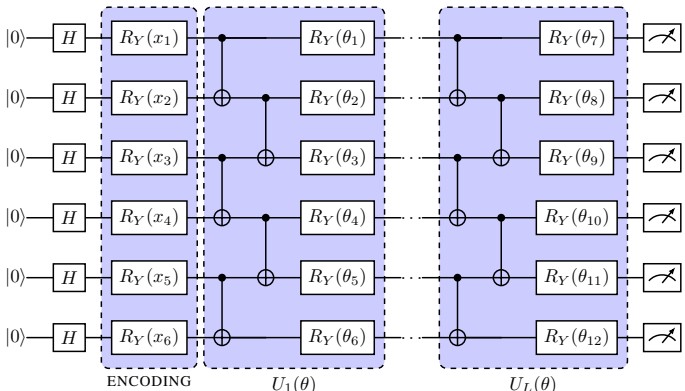

Figure 2: The ansatz used in our work, which consists of a unitary angle encoding followed by $L = 2$ trainable variational layers, which in turn consist each of entangling and parameterized $RY$ rotations for each qubit.

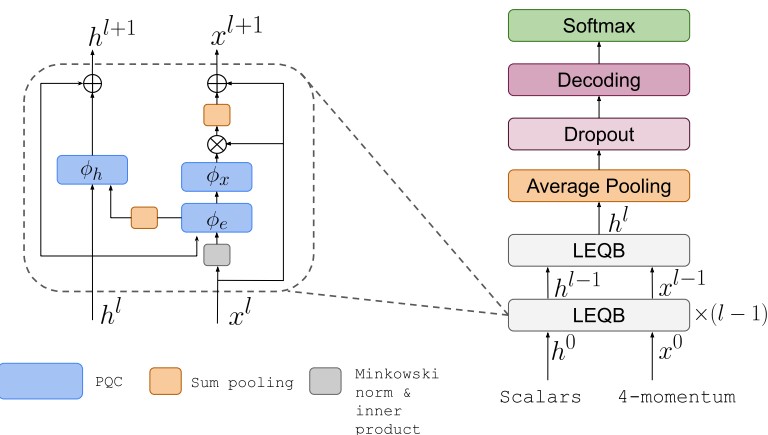

Figure 3: Lorentz-Equivariant Quantum Block (LEQB). This block ensures equivariance in the coordinate update function $x_i^{l+1}$ and invariance in the scalar update function $h_i^{l+1}$.

### 4.4.1 Lie Equivariant Quantum Block

Lorentz Equivariant Quantum Block (LEQB) is the main piece of our model (see Fig. 3). We aim to fundamentally learn deeper quantum representations of $x^l$ and $h^l$,

$$x_i^{l+1} = x_i^l + c \sum_{i=1}^{n} \phi_x(m_{ij}^l)x_j^l, \tag{2}$$

$$h_i^{l+1} = h_i^l + \phi_h(h_i^l, c \sum_{i=1}^{n} w_{ij}h_j^l), \tag{3}$$

where $x^l$ are the 4-momentum observables and $h^l$ are the particle scalars in the input layer $l = 0$, and $m_{ij}$ is the message between particles $i$ and $j$, defined by

$$m_{ij}^l = \phi_e\left(h_i^l, h_j^l, \psi\left(||x_i - x_j||^2\right), \psi\left(\langle x_i^l, x_j^l \rangle\right)\right), \tag{4}$$

where $||x_i - x_j||^2 = \eta(x_i - x_j, x_i - x_j)$ is the squared Minkowski norm, and $\langle x_i^l, x_j^l \rangle$ is the Minkowski inner product, defined by the bilinear form $\langle x_i, x_j \rangle = x_i^T \eta x_j$, where $\eta = diag(-1, 1, 1, 1)$ is the Minkowski metric. Finally, $\psi$ is just a normalization constant for large distributions to improve training stability, as defined in [25].

## 5 Experiments & results

After substituting $\phi_e$, $\phi_x$, $\phi_h$, and $\phi_m$ for parameterized circuits one at a time, we find the results (accuracy and losses over training epochs) shown in the first four panels in Figures 4 and 5. When all four modules $\phi_e$, $\phi_x$, $\phi_h$ and $\phi_m$ are represented with quantum circuits, we find the results shown in the lower left panels in Figures 4 and 5. Finally, the results in the lower right panels correspond to the fully classical model LorentzNet [25]. In all experiments, we used *AdamW* [43] as our optimizer with a learning rate $\gamma = 10^{-3}$ and weight decay of $\lambda = 10^{-2}$. Our simulations are idealized in the sense that we ignore any noise and assume an infinite number of shots. For warming up, we used a combination of cosine annealing [44] and gradual warm-up [45]. The total number of trainable parameters in all models can be seen in Table 1.

| Model | $\phi_e$ | $\phi_h$ | $\phi_m$ | $\phi_x$ | full_quantum | LorentzNet |
|---|---|---|---|---|---|---|
| Trainable Parameters | 668 | 1100 | 1090 | 998 | 592 | 1088 |

Table 1: The number of trainable parameters for the different models.

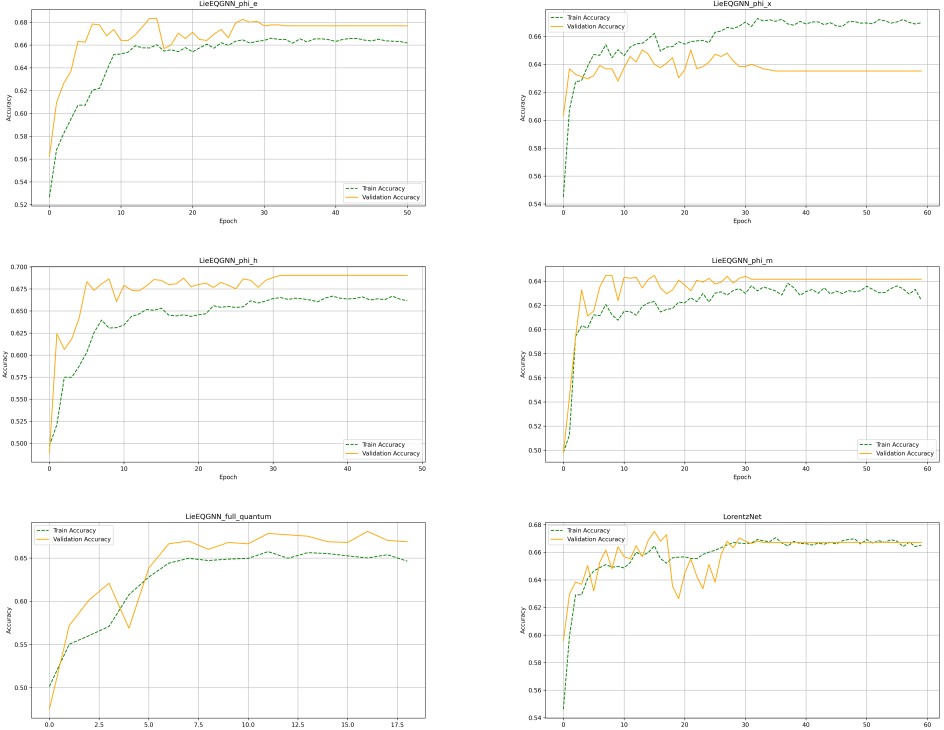

Figure 4: Classification accuracies for LieEQGNN with quantum architectures in place of $\phi_e$ (top left), $\phi_x$ (top right), $\phi_h$ (middle left), $\phi_m$ (middle right). The lower left panel shows the result from the model where all four $\phi_e$, $\phi_x$, $\phi_h$ and $\phi_m$ are represented with quantum circuits. The lower right panel is the result from the classical model LorentzNet [25].

## 6    Conclusion

Our study demonstrates that the performance of Lie-EQNNs is comparable to or slightly better than that of their classical counterparts, LorentzNet. This highlights the potential of quantum-inspired architectures in resource-constrained settings. The code used for this study is publicly available at `https://github.com/ML4SCI/QMLHEP/tree/main/Lie_EQGNN_for_HEP_Jogi_Suda_Neto`.

## Acknowledgments

This research used resources of the National Energy Research Scientific Computing Center, a DOE Office of Science User Facility supported by the Office of Science of the U.S. Department of Energy under Contract No. DE-AC02-05CH11231 using NERSC award NERSC DDR-ERCAP0025759. SG is supported in part by the U.S. Department of Energy (DOE) under Award No. DE-SC0012447. KM is supported in part by the U.S. DOE award number DE-SC0022148. KK is supported in part by US DOE DE-SC0024407.

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
