# OpenReview forum: "Lie-Equivariant Quantum Graph Neural Networks"
_NeurIPS.cc/2024/Workshop/MLNCP — MLNCP Poster_

### Official Review · Reviewer_vhcZ · 2024-10-07
**Good motivation and methodology.  Probably a decent incremental result.**

**Rating:** 6
**Confidence:** 2

**Review:**

This paper is about symmetric NNs, applied to HEP problems involving LHC data.  The motivation here is the use of GNNs for performing inference on this data, and the potential of using quantum GNNs instead of classical ones, with the appropriate symmetries (Lie equivariance).

A few criticisms:
* Is there a reference for the concept of quantum GNNs, or is this paper the first one to do so?  Previous quantum GNN papers were not cited, to my knowledge.
* Why is this related to an Ising model?  I know what an Ising model is, but that does not usually come up in the context of GNNs.  I am wondering whether they are using the Ising model just because that is one of the things that quantum computers can easily simulate.
* What is LEQB, the thing in Fig. 2?  Unclear.
* Where was this experiment run, on a physical quantum computer or a simulator?  If so, which?

Good they make their code public.

Evaluation: probably a publishable incremental result.  30-50th percentile.

---

### Official Review · Reviewer_6H9A · 2024-10-09
**The authors choose a promising task and choose a well-motivated circuit design with reasonable classical comparisons.  However, more work needs to be done to motivate the significance of this result, as well as clarify some vague points in the paper**

**Rating:** 4
**Confidence:** 4

**Review:**

The authors decide to apply quantum machine learning to a problem from particle physics - a source of physical data for which a quantum-based architecture might well be naturally suited. Further, they take the novel inspiration from geometric deep learning to choose the inductive bias of their quantum circuit to respect the symmetry of the Lorentz group. With commendable due diligence, the authors also compare their results to a classical neural network that is structured to respect the same symmetry.

Major Comments:

One of the authors' key results is that quantum networks (LEQNN's) empirically appear to perform about as well as LorentzNet. An explanation of the significance of this is missing. Is achieving comparable performance to LorentzNet something that is very difficult for most classical architectures? Is the fact that it performs even equally well surprising? If so, the proper context needs to be provided to justify this.
Further, it is unclear that comparable or marginally better performance is sufficient to motivate this, considering the additional costs associated with training quantum neural networks - hardware noise, shot noise, as well as the efficiency of gradient-based training for the quantum neural network (e.g. only known case of general backprop scaling in gradient computation requires multiple copies of the circuit whereas this can be easily implemented digitally -important considering the number of parameters!). This is to say nothing for the costs of having a quantum computer on hand to begin with. Therefore, at least *some* empirical evidence  is needed (perhaps numerics with deeper circuits) that significantly better performance is possible (say, 80% vs 67% accuracy), without encountering issues such as barren plateaus, for quantum circuits over LorentzNets with similar depths.

Additionally, given the marginal improvements seen, it's important to clarify: did the training of the quantum system assume a noiseless implementation in the infinite shot limit? Once the shot noise, or even hardware noise, of NISQ devices is approximated, how does this influence results?

Finally, I think it would be reasonable to leave the door open to negative results, which are comparably valuable. In particular, is there a good reason to suppose that a quantum system with this architecture will always perform, at best, only marginally better than something like LorentzNet (perhaps the latter already captures most of the relevant info)?


Minor Comments:

While the authors list the number of parameters used, it is unclear to me how many layers are used in their experiments.  From Figure 2, it seems that there are only two layers - but such a circuit is shallow enough to be easily classically simulable for any number of qubits, given that they seem to measure *local* observables. My instinct would be to divide the number of parameters by the number of qubits...but it seems that a measurement takes place immediately after the two layers are applied? Do the authors not reset between measurements, and that's how additional layers are applied? I would ask for clarification on this, both in the text and with a clearer image for layers.


The authors go into quite a bit of detail on the virtues on lie-equivariant neural networks, but don't really go into much detail as to why their quantum circuit architecture is equivariant under the Lorentz group. I would request adding a section elaborating on this (perhaps in lieu of the current section on quantum graph neural networks, which seems largely tangential to the circuit architecture presented).

---

### Decision · Program_Chairs · 2024-10-10

Accept (Poster)